# HIF-PH Encoded by *EGLN1* Is a Potential Therapeutic Target for Chronic Lymphocytic Leukemia

**DOI:** 10.3390/ph15060734

**Published:** 2022-06-10

**Authors:** Wancheng Guo, Daomiao Liang, Peilong Wang, Le Yin, Huifang Zhang, Cheng Xing, Zineng Huang, Yinghua Wu, Heng Li, Zhao Cheng, Xiaojuan Xiao, Jing Liu, Zhihua Wang, Hongling Peng

**Affiliations:** 1Department of Hematology, The Second Xiangya Hospital, Molecular Biology Research Center, School of Life Sciences, Hunan Province Key Laboratory of Basic and Applied Hematology, Central South University, Changsha 410011, China; 2204160215@csu.edu.cn (W.G.); 198211044@csu.edu.cn (P.W.); sammyyl@163.com (L.Y.); z2204140328@csu.edu.cn (H.Z.); xingcheng@csu.edu.cn (C.X.); zineng.huangchn@gmail.com (Z.H.); liheng001@csu.edu.cn (H.L.); chengzhao@csu.edu.cn (Z.C.); xiaojuan_xiao@csu.edu.cn (X.X.); liujing2018@csu.edu.cn (J.L.); 2Hunan Engineering Research Center of Cell Immunotherapy for Hematopoietic Malignancies, Changsha 410011, China; 3Xiangya School of Medicine, Central South University, Changsha 410013, China; 8301180212@csu.edu.cn; 4The First Affiliated Hospital of Hunan Normal University (Hunan Provincial People’s Hospital), Changsha 410005, China; liangdaomiao1997@163.com

**Keywords:** CLL, hypoxia-inducible factor prolyl hydroxylase (HIF-PH), molidustat, *EGLN1*

## Abstract

Owing to the recent emergence of drug resistance to Bruton’s tyrosine kinase inhibitors (BTK) in chronic lymphocytic leukemia (CLL) treatment, it is crucial to identify alternative therapeutic targets. Therefore, we aimed to identify therapeutic options for CLL besides BTK. We identified that HIF1A expression was higher in CLL patients than in controls, which may suggest good prognosis. We used a lentiviral knockdown of *EGLN1* (encoding hypoxia-inducible factor prolyl hydroxylase [HIF-PH]) and found that the growth of MEC-1 cells slowed in the knockdown group. Treatment of CLL cell lines MEC-1 and HG3 with the HIF-PH inhibitor molidustat showed that molidustat could induce apoptosis in a concentration-dependent manner in CLL cells and had low cytotoxicity at this concentration. *CXCR4*, *HIF1A*, *SLC2AI*, and *VEGF*, the downstream molecules of the HIF pathway, were upregulated after molidustat treatment. Western blotting results indicated that molidustat increased HIF1A expression in CLL cell lines and cells from CLL patients, and sequencing/quantitative PCR analysis demonstrated that the ribosome biogenesis pathway was inhibited in MEC-1 cells after molidustat treatment. We further identified synergistic cytotoxicity of molidustat in combination with ibrutinib on the MEC-1 and HG3 cell lines at certain concentrations. Therefore, molidustat is a potential therapeutic option for CLL.

## 1. Introduction

Chronic lymphocytic leukemia (CLL) is a malignant B cell proliferative disease. As the most common type of leukemia in western countries, CLL has a lifetime probability of approximately 0.6% [1], usually developing in middle and old age [2]. It is currently thought that recurrent genetic lesions are associated with CLL pathogenesis and/or disease progression. B cell antigen receptor (BCR) signaling and the leukemic microenvironment of CLL also play an important role in CLL disease progression [3]. Treatments for CLL are varied, and conventional chemotherapeutic agents include alkylating agents (bendamustine) and purine analogs (fludarabine). In recent years, the clinical application of anti-CD20 monoclonal antibodies and B cell signaling pathway inhibitors has led to more options for patients with CLL [4]. Currently, targeted inhibitors in clinical use are divided into three main categories: Bruton’s tyrosine kinase (BTK) inhibitors (ibrutinib), B cell lymphoma 2 inhibitors (venetoclax), and phosphatidylinositol-3-kinase inhibitors (idelalisib). Among them, the development and clinical application of a targeted inhibitor, i.e., ibrutinib, developed based on the BCR pathway, has greatly prolonged the progression-free survival of CLL patients [5].

Targeted inhibitors have relatively few side effects, and their use in the treatment of CLL has continuously increased. However, there is the possibility of development of drug resistance due to the long-term use of targeted inhibitors [6]. The causes of BTK resistance primarily include genetic mutations, e.g., the C481S mutation in BTK [7] and the PLCG2 mutation [8], as well as activation of B cell proliferation pathways other than BCR [9]. Therefore, identification of the cause of drug resistance and new therapeutic targets for CLL are pivotal. Currently, several promising targets have been identified, including ROR1 [10], BAFF-R [11], and SYK and LYN, two molecules upstream of BTK [12].

Hypoxia-inducible factor-1 (HIF-1) is ubiquitous in human and mammalian cells and rapidly degraded by the intracellular oxygen-dependent ubiquitin protease degradation pathway under normoxic conditions. HIF-1 is only stably expressed under hypoxic conditions. HIF pathway activation can cause changes in the expression of many genes, such as vascular endothelial growth factor (*VEGF*), *CXCR4*, and *SLC2AI*. In recent years, HIF-1α has been shown to regulate the expression of chemokine receptors and cell adhesion molecules in CLL cells that control the interaction of leukemic cells with the bone marrow and spleen microenvironment. Inactivation of HIF-1α impairs chemotaxis and cell adhesion to the matrix, reduces bone marrow and spleen colonization in xenograft and allograft CLL mouse models, and prolongs survival [13]. HIF-1α is overexpressed in CLL cells in TP53-disrupted patients, and the HIF-1α inhibitor BAY87-2243 exerts cytotoxic effects on CLL cells and has shown antitumor activity in Eμ-TCL1 mice [14]. These lines of evidence suggest HIF-1α as a promising therapeutic target for CLL. However, leukemic cells are well known to be present in the blood, which is rich in oxygen compared with other tissues, and usually not considered in the context of tumor angiogenesis. Therefore, the mechanism by which targeted inhibition of HIF leads to CLL cell death remains to be investigated.

HIF-PH is an enzyme that degrades HIF1A protein and is encoded by the *EGLN1* gene. Inhibition of the HIF-PH protein stabilizes the HIF1A protein in cells. Molidustat is a targeted inhibitor of HIF-PH whose primarily clinical use is in the treatment of renal anemia [15]. Current studies have shown that molidustat induces S/G2 cell cycle arrest in breast cancer cells and inhibits breast cancer cell proliferation in vivo [16]. However, the effect of molidustat in other solid tumor leukemias has not been studied. Figure 1 shows the chemical structure of molidustat and other CLL targeted drugs mentioned above.

In this study, we found that HIF-PH encoded by EGLN1 is a potential therapeutic target for chronic lymphocytic leukemia and its inhibitor, molidustat, has considerable killing effect on CLL cell lines. Figure 2 displays the workflow of this article.

## 2. Results

### 2.1. GEO Database Analysis of the Expression, Outcomes, and Functional Characteristics of the HIF1A in CLL

Valsecchi et al. identified HIF1A as a promising therapeutic target for CLL [13]. To investigate the role of *HIF1A* expression in CLL and its outcomes, we analyzed the GSE50006 and GSE22762 datasets from the GEO database. We performed differential analysis of gene expression between CLL patients and normal subjects in GSE50006. Figure 3A,B show a heat map and volcano plot of patient gene expression (top 100) from the GSE50006 database, respectively. Interestingly, comparison of transcriptome data between CLL patients and normal individuals revealed that HIF1A protein expression was lower in CLL patients, whereas the expression of its antagonist, HIF1AN, was higher in CLL patients (Figure 3C,D). Survival analysis of the GSE22762 dataset showed that high *HIF1A* expression may indicate good prognosis for CLL patients (Figure 3F). Next, we performed gene ontology (GO) and Kyoto Encyclopedia of Genes and Genomes (KEGG) analyses of differential genes (top 2000) between *EGLN1* knockdown and controls in the GSE77173 dataset and found that positive regulation of cytokine production and T cell receptor signaling pathway were enriched (Figure 3G).

### 2.2. Knockdown of EGLN1, a Gene Encoding a HIF1A-Degrading Protein HIF-PH, Resulted in Apoptosis of CLL Cells

To further explain the good prognosis of CLL patients in the database with high expression of HIF1A, we focused on HIF-PH, a key protein of the HIF1A protein degradation pathway. We used shRNA to construct a lentivirus to knock down *EGLN1*, which encodes HIF-PH, in CLL cell lines. The control group was transfected with lentivirus containing the same vector with a scrambled sequence. After transfection, the efficiency of individual lentiviral knockdown was determined using qPCR (Figure 4A). After 3 days of puromycin screening, plates were seeded at a density of 10^5^/mL and the growth curves of each group were plotted. We found that cells of the knockdown group grew significantly slower than those of the control group (Figure 4B). Furthermore, we used flow cytometry to detect apoptosis in each group and found that the apoptosis rate of the knockdown group was higher than that of the control group (Figure 4C).

### 2.3. Molidustat Inhibited Proliferation and Promoted Apoptosis in CLL Cells

To further investigate the value of targeted inhibition of HIF-PH, we treated the CLL cell lines MEC-1 and HG3 and normal human mononuclear cells with the HIF-PH-targeting inhibitor molidustat. The results showed that the half maximal inhibitory concentration (IC_50_) of molidustat on MEC-1 and HG3 was approximately 20 μM (Figure 5A,B), but molidustat had little effect on normal human mononuclear cells at this concentration (Figure 3C). Next, we examined the effect of molidustat on apoptosis in MEC-1 cells using flow cytometry, with ibrutinib as a positive reference. We found that molidustat induced apoptosis in MEC-1 cells in a concentration-dependent manner (Figure 5D).

### 2.4. Molidustat Remains Effective against Ibrutinib-Resistant MEC-1 Cells

Ibrutinib resistance is a major challenge in the treatment of CLL [17]. To further investigate whether molidustat has the same inhibitory effects on ibrutinib-resistant strains of CLL, we intermittently treated MEC-1 with low concentrations of ibrutinib and gradually increased ibrutinib concentrations to induce ibrutinib resistance in MEC-1 cells [18]. After treatment with different concentrations of molidustat, we found that approximately 30% of ibrutinib-resistant MEC-1 cells survived treatment with 40 μM ibrutinib (Figure 6A), whereas almost all drug-resistant MEC-1 cells were killed by 40 μM molidustat (Figure 6B).

### 2.5. Molidustat Increased HIF1A Protein Expression and Induced Downstream Activation in CLL Cell Lines

To verify whether molidustat induces molecular changes by inhibiting HIF-PH, we treated MEC-1 cells with molidustat and determined HIF1A protein expression using Western blotting. We found that molidustat increased HIF1A protein expression in MEC-1 cells in a concentration-dependent manner (Figure 7A). Primary CLL cells were treated with 0 or 20 μM molidustat, and HIF1A protein expression levels were found to be elevated after molidustat treatment (Figure 7B). Next, we examined the expression of *HIF1A* and downstream molecules using qPCR and found that the mRNA levels of *HIF1A* and several downstream molecules (*CXCR4*, *SLC2AI* and *VEGF*) were significantly elevated after molidustat treatment (Figure 7C–F).

### 2.6. Molidustat Inhibited Ribosome Biogenesis and Increased TP53I Family Gene Expression in CLL Cells

To further investigate the mechanism of molidustat inhibition of CLL cell proliferation, we examined the mRNA expression levels in the molidustat-treated and control groups using second-generation sequencing technology. We performed enrichment analysis of mRNAs of differential genes in molidustat-treated and control groups (GO analysis results are shown in Figure 8A and KEGG analysis results are shown in Figure 8B). We found that the ribosome biogenesis pathway was enriched after molidustat treatment (Figure 8C). In addition, we found that the expression of oncogenes such as *TP53I* and *TP53INP* was upregulated in the molidustat-treated group compared with control. Existing studies have shown that TP53I and TP53INP can inhibit cell proliferation through various pathways [19,20] and that inhibition of the ribosome biogenesis pathway is responsible for the slowed proliferation of tumor cells [21]. We verified the gene expression changes in transcriptome sequencing using qPCR and found that the expression levels of ribosome biogenesis pathway-related genes (*MDN1*, *NOL6*, *POP1*, and *WDR3*) decreased (Figure 8D), whereas the expression levels of the *TP53I13*, *TP53I3*, *TP53INP1*, and *TP53INP2* genes increased (Figure 8E), which is consistent with transcriptome sequencing results.

### 2.7. Synergistic Effects of Molidustat and Ibrutinib on CLL Cells

Based on previous results, we found that the effect of molidustat on ibrutinib-resistant MEC-1 did not differ greatly from that on wild-type MEC-1 cells, and that molidustat inhibited the ribosome biogenesis pathway and promoted *TP53I*-related gene expression in MEC-1 cells. Molidustat is an inhibitor of HIF-PH with a different mechanism of action from ibrutinib, which inhibits BTK. To investigate whether the two have synergistic cytotoxicity on CLL cell lines and whether their combination has synergistic effects, we treated MEC-1 and HG3 cells with different combinations of ibrutinib and molidustat at different concentrations (Figure 9A,B). As a result, the two drugs showed synergistic cytotoxicity in CLL cell lines. The combination of 2.5 μM molidustat and 0.3125 μM ibrutinib had a synergistic index of 0.45 in MEC-1 cells, and the combination of 2.5 μM molidustat and 0.625 μM ibrutinib had a synergistic index of 0.41 in HG3 cells.

## 3. Discussion

Various treatments are currently available for CLL, especially after the development and clinical use of targeted drugs (ibrutinib); therefore, the disease prognosis has greatly improved. However, BTK resistance in CLL remains a challenging clinical problem; thus, novel therapeutic targets and targeted drugs for CLL are necessary.

Many molecules have already been identified as candidate targets for CLL, such as ROR1 [10], BAFF-R [11], and upstream molecules BTK, SYK, and LYN [12]. HIF1α, which was recently discovered, may also serve as a possible therapeutic target for CLL [22].

The function of the *HIF1A* gene in hematologic tumors has been controversial. In solid tumors, hypoxia occurs in the local microenvironment due to the high rates of tumor cell proliferation and metabolism. Hypoxia can inhibit the function of HIF-PH and activate the HIF1A pathway, which increases the expression of molecules such as VEGF and promotes tumor angiogenesis [23,24]. However, in hematologic tumors, tumor cells are present in blood vessels and are directly in contact with the blood, which serves as a microenvironment and not as a fluid tissue. This is likely the reason why many current HIF1A inhibitors have varying effects on hematologic tumors [25]. *HIF-1α* knockdown has been shown to reduce bone marrow homing of MEC-1 cells. The HIF-1α inhibitor EZN-2208 reduces mRNA levels of HIF-1α target genes and has a significant inhibitory effect on CLL cells [13]. HIF-1α is overexpressed in leukemic cells in TP53-disrupted patients, and the selective HIF-1α inhibitor BAY87-2243 has been shown to have antitumor activity in CLL [14]. Interestingly, our mRNA sequencing results based on a large-sample public database (GSE50006) showed that mRNA expression levels of *HIF1A* were higher than normal in CLL, whereas mRNA expression levels of *HIF1AN* were lower than normal. Survival analysis of the GSE22762 dataset showed that high HIF1A expression may suggest a good prognosis for CLL patients. Therefore, the specific role played by HIF1A in patients must be further investigated. We focused on HIF-PH, a protein that degrades HIF-1α, and knocked down *EGLN1*, an important gene encoding the HIF-PH protein, in the CLL cell line MEC-1 and found that the cell growth rate substantially slowed, and the apoptosis rate increased.

Inhibition of HIF-PH by molidustat, which is used for the treatment of renal anemia, leads to activation of the HIF pathway and increased expression of erythropoietin downstream of the HIF pathway, thereby improving anemia [26]. Kachamakova-Trojanowska et al. treated cells with molidustat and found increased protein expression of HIF-1α and its downstream factor VEGF. Molidustat can induce cell cycle arrest in breast cancer cells through the p53 pathway [16]. We examined the expression of oncogenes from the *TP53I* and *TP53INP* families in MEC-1 cells after molidustat treatment and found that the expression levels of *TP53I13*, *TP53I3*, *TP53INP1*, and *TP53INP2* remarkably increased, which may lead to the death of CLL cells.

Ribosomes tend to be highly expressed in tumor cells and are used to maintain cell growth and division. Alterations in ribosomal RNA synthesis and dysregulation of certain ribosomal proteins are associated with tumor progression [27]. Altered expression of ribosomal proteins may affect ribosome biogenesis, leading to a condition known as nucleolar or ribosomal stress. Cytoplasmic ribosomal proteins (CRPs) can be free in the cytoplasm or translocate to the nucleus to perform their functions. Some of these free CRPs act through the p53-MDN2 pathway; other CRPs act through a different pathway that is independent of p53 [28]. Currently, the 5S ribonucleoprotein (RNP)-MDM pathway is thought to be associated with chemotherapy agent-induced nucleolar stress responses, nutritional deficiencies, overexpression of the tumor suppressor p14ARF, and responses to different forms of stress, including hypoxia and oxidative stress. In addition, 5S RNP is involved in the induction of p53-mediated cellular senescence in response to oncogenic and replicative stress [29]. Ribosomal proteins regulate the cell cycle, DNA repair, maintain genomic integrity, cell proliferation, apoptosis, autophagy, cell migration, and invasion [30]. In the current study, we found that the ribosome biogenesis pathway was inhibited through enrichment analysis of mRNA expression in MEC-1 cells before and after molidustat treatment. We further verified the expression of *MDN1*, *NOL6*, *POP1*, and *WDR3* in the ribosome biogenesis pathway using qPCR, and found that molidustat could reduce the expression of these genes.

## 4. Materials and Methods

### 4.1. Cell Lines and Materials

The human CLL cell line MEC-1 was purchased from Cobioer Biosciences Co. (Nanjing, China). The HG3 cell line was provided by Prof. Kailin Xu, Xuzhou Medical College. HS-5 bone marrow stromal cells were provided by Prof. Jing Liu, College of Life Sciences, Central South University. 293-T cells were purchased from American Type Culture Collection (Manassas, VA, USA). MEC-1 cells were cultured in Iscove′s modified Dulbecco′s medium (IMDM) (Gibico) containing 10% fetal bovine serum, HG3 cells were cultured in RPMI 1640 medium containing 10% FBS, and HS-5 and 293-T cells were cultured in Dulbecco’s modified Eagle medium (DMEM) (Gibico) containing 10% FBS. All cells were incubated in an incubator containing 95% air and 5% CO_2_ at 37 °C (Thermo Scientific, Waltham, MA, USA) and tested regularly for mycoplasma infection using a mycoplasma detection kit (Vazyme, Nanjing, China).

### 4.2. Bioinformatics Analysis

Gene expression values between CLL patients and normal controls in the GSE50006 dataset were analyzed using R v.4.0.2 software (limma package) to identify differentially expressed genes. The survival and survminer packages were used to perform survival analysis of the GSE22762 dataset using median expression values of *HIF1A* and *HIF1AN* to determine the level of gene expression. The Wilcoxon test was used to compare the expression values of the *HIF1A* and *HIF1AN* genes in the CLL and normal samples. GO and KEGG analyses of the differentially expressed genes were performed using the clusterProfiler package. The KEGG Gene Set Enrichment Analysis (GSEA) database (https://www.gsea-msigdb.org/gsea/downloads.jsp, accessed on 1 November 2021) and enrichplot package were used for GSEA enrichment analysis.

### 4.3. Virus Preparation and Transfection

The PLKO.1 plasmid (from Zhijian Qian, University of Florida, Gainesville, FL, USA) was used as a vector and linearized by double digestion using *Eco*RI and *Bst*HI. Next, the linearized vector was separated by agarose gel electrophoresis and recovered using a gel recovery kit (E.Z.N.A. Gel Extraction Kit, Omega Bio-Tek, Norcross, GA, USA). The control scramble sequences and shRNA primer sequences are shown in Appendix A and were synthesized by Tsingke Biotechnology Co., Ltd. (Beijing, China). Next, shRNA was ligated to the linear vector using T4 ligase (Tsingke Biotechnology Co., Ltd., Beijing, China) and the ligated product was transformed into 293-T cells (purchased from American type culture collection). Next, the competent cells were amplified by shaking, plated, and then single bacterial clones were picked. Detection of the plasmid sequence in single clones was performed by Tsingke Biotechnology Co., Ltd. SnapGene 5.2 software was used to compare the shRNA sequences with the sequences of the ligated plasmids. After confirming that the clone carried the correct target plasmid, it was amplified by shaking, and the target plasmid was extracted using the E.Z.N.A. Plasmid DNA Mini Kit I (Omega Bio-Tek, Norcross, GA, USA). The other two packaging vectors of the virus were PxpaX2 and PMD2g (from Zhijian Qian, University of Florida). Next, the three plasmids (ratio: 2 μg:1 μg:3 μg) were added to adherent 293T cells. At the same time, polyethylenimine (purchased from Thermo Fisher Scientific, Waltham, MA, USA) was added to facilitate plasmid entry into the 293T cells. After 4 h, the culture medium of the 293T cells was replaced. Next, the 293T culture medium was changed every 24 h and the clear viral suspension was collected for seven consecutive days.

The viral supernatant was centrifuged (1650× *g*) to remove 293T cells and cell debris, after which the viral suspension was added to MEC-1 cells. Polybrene was added to improve the efficiency of virus infection. Cells were centrifuged on a plate centrifuge for 3 h (500× *g*), after which the MEC-1 culture medium was changed every 24 h; the transfection was repeated three times. After the last transfection, cells were cultured with normal medium for 12 h, followed by changing to IMDM medium containing 2 μg/mL puromycin to screen the cells. After 72 h of screening, the target gene *EGLN1* knockdown efficiency was determined using qPCR, and growth curve analysis and apoptosis assays were performed.

### 4.4. Growth Curve of Cells after Transfection

The initial density of cells was 10^5^/mL and the total system volume was 1 mL. Every 48 h, the cells were counted again, and the culture medium was replaced. When the cell density exceeded 7 × 10^5^/mL, all cells were added to 1 mL of medium until the seventh day.

### 4.5. Treatments

The small molecules molidustat and ibrutinib used in the study were purchased from TopScience Biotech (Shanghai, China). Dimethyl sulfoxide (DMSO) was purchased from Beyotime Biotechnology (Shanghai, China). Molidustat and ibrutinib stock solutions (10 mM) were obtained by dissolving in DMSO.

### 4.6. Cell Viability

The viability of cells treated with molidustat or control cells treated with an equal amount of DMSO vehicle was determined using the cell counting kit-8 (CCK-8) assay (Beyotime Biotechnology, Shanghai, China). Briefly, MEC-1 and HG3 cells were cultured in 100 μL complete medium at a density of 4 × 10^4^ cells/well in 96-well plates, and the experimental group was treated with different concentrations of molidustat, and the control group was treated with equal amounts of DMSO vehicle for 48 h. Next, 10 μL of CCK-8 reagent was added and the cells were incubated for 2 h at 37 °C. The absorbance was measured at 450 nm (OD_450_) using a spectrophotometer (Thermo Electron, Waltham, MA, USA) and the cell viability was calculated.

### 4.7. Construction of Ibrutinib-Resistant Strains

We constructed the MEC-1-induced ibrutinib-resistant strain using the method described by Kapoor et al. [18]. MEC-1 cell density was controlled to 5–7 × 10^5^ w/mL, initially incubated with ibrutinib at one-fifth of the IC_50_ (approximately 0.5 μM), incubated for 2 days, and then switched to ibrutinib-free IMDM medium for 2 days, after which the concentration of ibrutinib was increased. This was repeated until the final ibrutinib concentration was increased to 20 μM.

### 4.8. Apoptosis Assays

MEC-1 cells were first seeded in 6-well plates at a density of 4 × 105/mL and treated with 0, 10, 20, or 40 μM of molidustat for 48 h, ensuring maintenance of the same concentration of DMSO in every group. After drug treatment for 48 h, we performed apoptosis assays using Annexin-V-FITC kit (4A Biotech Co., Ltd., Beijing, China.) Cells were collected and washed in phosphate-buffered saline (PBS) (KeyGEN BioTECH Corp., Ltd., Nanjing, China), after which they were resuspended in annexin V buffer, incubated with annexin V antibody for 30 min away from light, and stained with propidium iodide dye. Finally, the percentage of apoptotic cells in each group was determined using flow cytometry (Becton, Dickinson and Company, New York, NY, USA).

### 4.9. RNA Extraction and Real-Time qPCR

Total RNA was extracted from MEC-1 cells using TRIzol reagent (Takara, Ōtsu City, Japan) and tested for purity and concentration using a NanoDrop 2000 spectrophotometer (Thermo Fisher Scientific, Waltham, MA, USA). DNA was removed from the RNA using a gDNA wiper mix (Vazyme, Nanjing, China), after which total RNA was reverse transcribed into cDNA using a cDNA reverse transcription kit (Vazyme). A fluorescence quantitative PCR kit was purchased from Vazyme. The final results were determined using the CFX96 Touch System (Bio-Rad Laboratories, Hercules, CA, USA), a real-time fluorescence quantitative PCR instrument. The expression of the target genes was normalized using β-actin expression, and the final difference between the experimental and control groups of the target gene was calculated; 2^−a^ indicates the fold change in gene expression. The specific sequences of the target gene primers are shown in Appendix A.

### 4.10. Western Blotting

Cells were collected by centrifugation, washed with PBS, lysed using radioimmunoprecipitation buffer (with phenylmethylsulfonyl fluoride (Thermo Fisher) and protease inhibitor (Roche) added), incubated on ice for 30 min, and then lysed using a sonicator (Sonics, Newtown, CT, USA). Cells were centrifuged for 15 min at 4 °C and the precipitate was discarded. Next, a loading buffer (BioNTech, Mainz, Germany) was added to the protein samples, and the samples were boiled for 10 min. The proteins were separated by electrophoresis in 10% sodium dodecyl sulfate-polyacrylamide gels (electrophoresis instrument purchased from Bio-Rad) and then transferred to a 0.45-μm polyvinylidene fluoride (PVDF) membrane (transfer apparatus purchased from Bio-Rad, PVDF membrane purchased from Millipore, Boston, MA, USA). Next, the membrane was blocked with 5% skim milk for 1 h, the milk was discarded, and the membrane was washed three times with PBS-Tween 20 (Sigma-Aldrich, St. Louis, MO, USA; 0.5 parts per thousand). Then, the membrane was blocked with anti-GAPDH and anti-HIF-1α antibodies diluted by primary antibody dilutions (Beyotime, Shanghai, China) and incubated overnight at 4 °C. Next, the membrane was washed three times with PBS + 0.05% Tween-20 (Sigma-Aldrich) and incubated with horseradish peroxidase (HRP)-conjugated secondary antibodies (Proteintech, Chicago, IL, USA) for 1 h at room temperature. Proteins were visualized using an HRP-based chemiluminescence kit (Invigentech, Irvine, CA, USA) and Gel Imaging System (Bio-Rad, Alfred Nobel Drive, Hercules, CA, USA).

### 4.11. Primary Cell Isolation, Culture, and Treatment

A peripheral blood single nuclei cell extraction kit (Solarbio, Beijing, China) was used to extract peripheral mononuclear cells from normal individuals and patients with CLL. As described by Seiffert et al. [31], mononuclear cells from patients with CLL were co-cultured with HS-5 cells for 48 h, followed by drug intervention for 48 h and protein and RNA extraction.

### 4.12. RNA Sequencing

MEC-1 cells in the control and experimental groups were treated with 20 μM and 0 μM molidustat for 48 h. Next, the cells were collected, washed three times with PBS, and lysed with TRIzol solution (Thermo Fisher Scientific). RNA sequencing and enrichment analysis were performed by Novogene (Beijing, China).

### 4.13. Analysis of Synergistic Effects

The Chou–Talalay Method and CalcuSyn software (Version1.0, ComboSyn, Paramus, NJ, USA) were used for drug synergy analysis. A synergistic index less than 1 suggested a synergistic cytotoxic effect of the two drugs on cells at those concentrations.

### 4.14. Statistics

GraphPad Prism 5.0 (GraphPad Software, San Diego, CA, USA) was used for statistical analysis. Analysis of variance was used to compare the differences between the experimental and control groups. *p* < 0.05 was considered statistically significant.

## 5. Conclusions

In conclusion, we conducted GEO-based large-sample differentially expressed gene and survival analyses and identified differences in *HIF1A* expression between CLL patients and normal controls as well as an effect on the survival of CLL patients. We focused on HIF-PH, a protein that degrades HIF1A. Our study suggests that the *EGLN1* gene may serve as a therapeutic target for CLL and that the HIF-PH inhibitor molidustat may inhibit CLL cell proliferation by inhibiting the ribosome biogenesis pathway.

## Figures and Tables

**Figure 1 pharmaceuticals-15-00734-f001:**
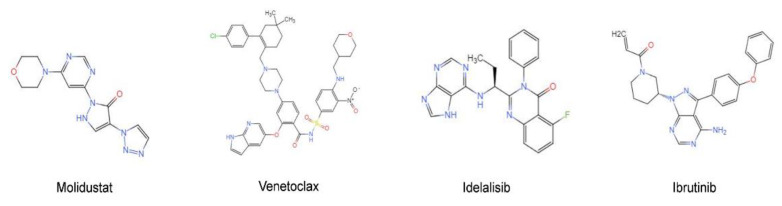
Chemical structure of Molidustat and other CLL targeted drugs.

**Figure 2 pharmaceuticals-15-00734-f002:**
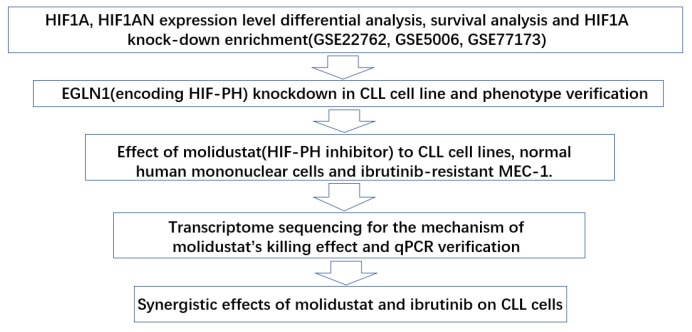
Flowchart of this work.

**Figure 3 pharmaceuticals-15-00734-f003:**
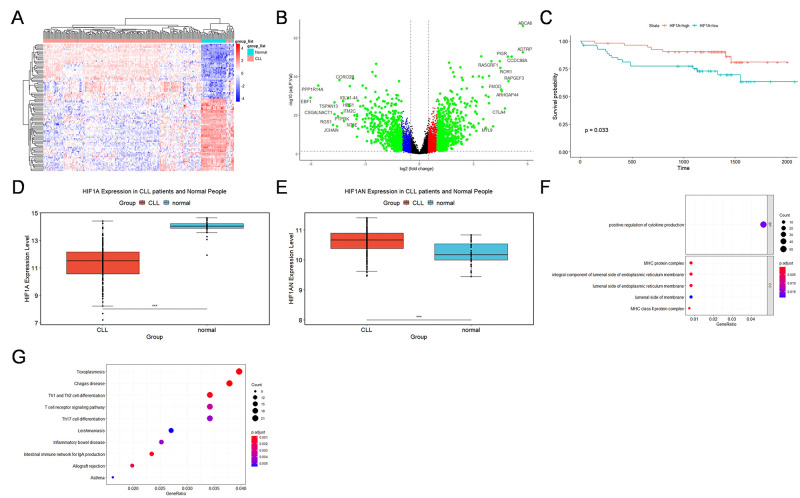
Single-gene analysis of *HIF1A* based on chronic lymphocytic leukemia (CLL) transcriptome data in the Gene Expression Omnibus (GEO) database. (**A**) Heat map of differential genes (top 100; dark red indicates larger logFC values and dark blue indicates smaller logFC values. (**B**) Volcano plot of differential gene analysis. Genes with logFC greater than 4 are labeled in the graph. (**C**) Survival analysis of the GSE22762 dataset showing high expression of HIF1A protein (high expression defined by median) indicates good prognosis for CLL patients. (**D**,**E**) Expression of HIF1A and HIF1AN proteins in CLL patients and normal controls. HIF1A protein levels were low in CLL patients and high in normal controls, and HIF1AN protein levels were high in CLL patients and low in normal controls (Wilcoxon test). (**F**) Gene ontology (GO) analysis of 2000 genes with the greatest difference between *EGLN1* knockdown and the control group. Positive regulation of cytokine production was enriched in biological process in GO analysis. (**G**) Kyoto Encyclopedia of Genes and Genomes (KEGG) analysis of the 2000 genes with the greatest difference between *EGLN1* knockdown and the control group. The T cell receptor signaling pathway was enriched (*** means *p* ≤ 0.001).

**Figure 4 pharmaceuticals-15-00734-f004:**
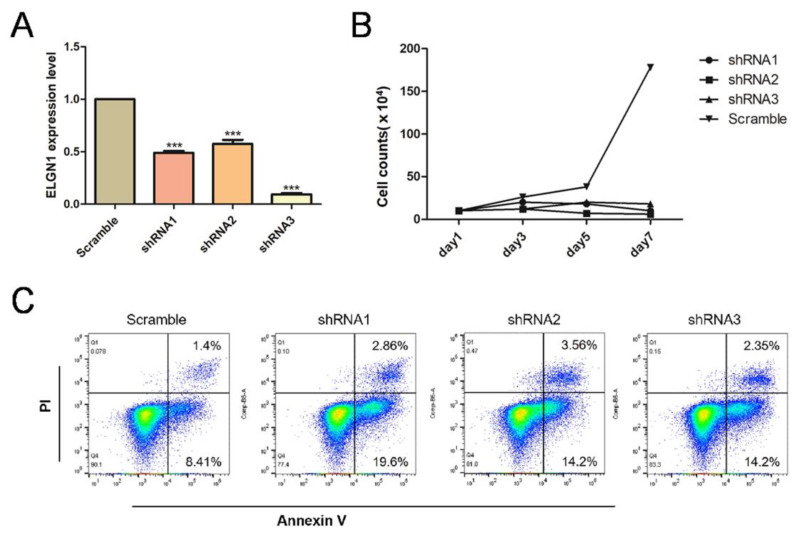
Effect of *EGLN1* knockdown on the growth of the CLL cell line MEC-1. (**A**) Quantitative PCR (qPCR) determination of the knockdown efficiency of the three short hairpin RNAs (shRNAs) on *EGLN1* gene knockdown in MEC-1 cells. All three shRNAs caused effective knockdown in MEC-1 cells (*** means *p* ≤ 0.001). (**B**) Growth curves of the shRNA knockdown group and the control group. The growth rate of the knockdown group was significantly lower than that of the control group. (**C**) Proportion of apoptotic cells in the knockdown and control groups. The percentage of early apoptotic cells in the knockdown group was higher than in the control group.

**Figure 5 pharmaceuticals-15-00734-f005:**
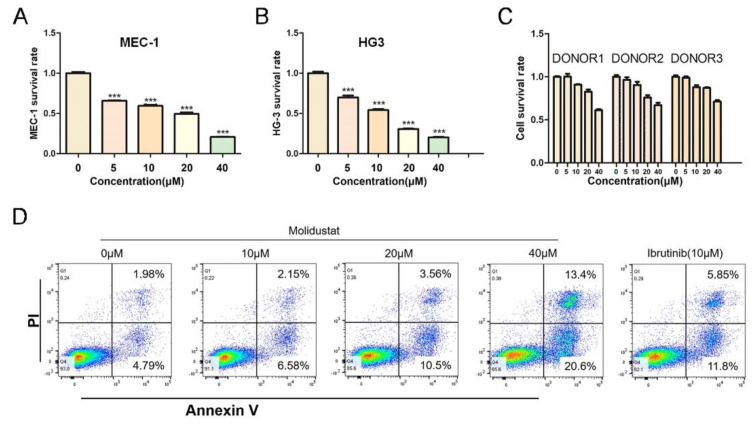
Effect of molidustat on MEC-1 and normal mononuclear cells. (**A**,**B**) CCK-8 assay of the effects of 0, 5, 10, 20, or 40 μM of molidustat on the MEC-1 and HG3 cell lines. Molidustat inhibited the proliferation of both cell lines in a concentration-dependent manner (*** means *p* ≤ 0.001). (**C**) CCK-8 assay of the effects of 0, 5, 10, 20, or 40 μM of molidustat on mononuclear cells in normal human peripheral blood. At these concentrations, molidustat had little effect on peripheral blood mononuclear cells. (**D**) MEC-1 cells were treated with 0, 10, 20, or 40 μM molidustat and 10 μM ibrutinib for 48 h, and the apoptosis rate was determined by flow cytometry. The apoptosis rate of MEC-1 cells increased as the concentration of molidustat was increased; 20 μM molidustat and 10 μM ibrutinib had comparable effects on the apoptosis rate of MEC-1 cells.

**Figure 6 pharmaceuticals-15-00734-f006:**
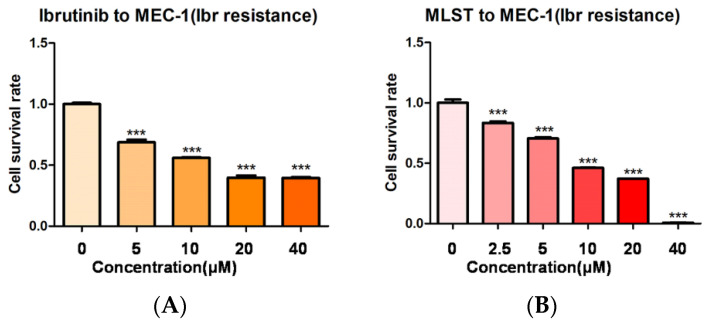
Effect of molidustat on ibrutinib-resistant MEC-1 cells. (**A**) Effect of different concentrations of ibrutinib on the survival of ibrutinib-resistant MEC-1 cells. A high concentration of 40 μM ibrutinib still resulted in 30% survival of ibrutinib-resistant MEC-1 cells (*** means *p* ≤ 0.001). (**B**) Cytotoxic effects of molidustat on ibrutinib-resistant MEC-1 cells. MEC-1 cells were treated with molidustat at the indicated concentrations for 48 h. CCK-8 results showed that 40 μM molidustat killed nearly all ibrutinib-resistant MEC-1 cells ( *** means *p* ≤ 0.001).

**Figure 7 pharmaceuticals-15-00734-f007:**
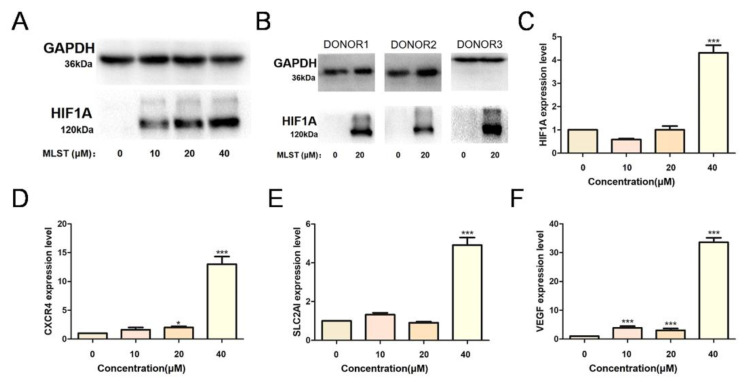
Molidustat increased HIF1A expression and induced hypoxia-inducible factor (HIF) pathway activation. (**A**) MEC-1 cells were treated with 0, 10, 20, or 40 μM molidustat. Western blotting results showed that molidustat stabilized HIF1α protein expression in MEC-1 cells in a concentration-dependent manner. (**B**) Primary cells from three CLL patients were treated with 0 or 20 μM molidustat for 48 h. Western blotting results showed that HIF1α protein was barely detectable in cells treated with 0 μM molidustat but was substantially increased after treatment with 20 μM molidustat. (**C**–**F**) MEC-1 cells were treated with 0, 10, 20, or 40 μM molidustat, and the expression of *HIF1A* and its downstream genes was determined using qPCR. The results showed that molidustat promoted the transcription of *HIF1A* and its downstream genes (* means *p* ≤ 0.05, *** means *p* ≤ 0.001).

**Figure 8 pharmaceuticals-15-00734-f008:**
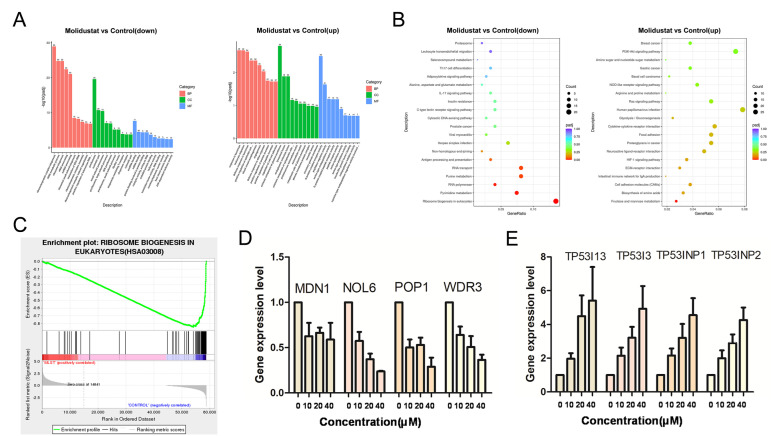
Possible mechanism of molidustat-induced apoptosis in MEC-1 cells. (**A**) MEC-1 cells were treated with 0 or 20 μM molidustat, and their RNA was extracted and sequenced. GO analysis was performed on the upregulated and downregulated differentially expressed genes of the MEC-1 cells in the experimental (20 μM) and control groups (0 μM). The histogram of enriched pathways is shown. (**B**) KEGG analysis was performed on the upregulated and downregulated differentially expressed genes, and the dot plot of KEGG enriched pathways is shown. (**C**) Gene Set Enrichment Analysis (GSEA) results of KEGG enrichment analysis of ribosome biogenesis showing that the majority of genes in the ribosome biogenesis pathway was downregulated after 20 μM molidustat treatment (most vertical lines distributed to the right of the horizontal axis). (**D**,**E**) MEC-1 cells were treated with 0, 10, 20, or 40 μM molidustat, and the expression of differential expressed genes was verified using qPCR.

**Figure 9 pharmaceuticals-15-00734-f009:**
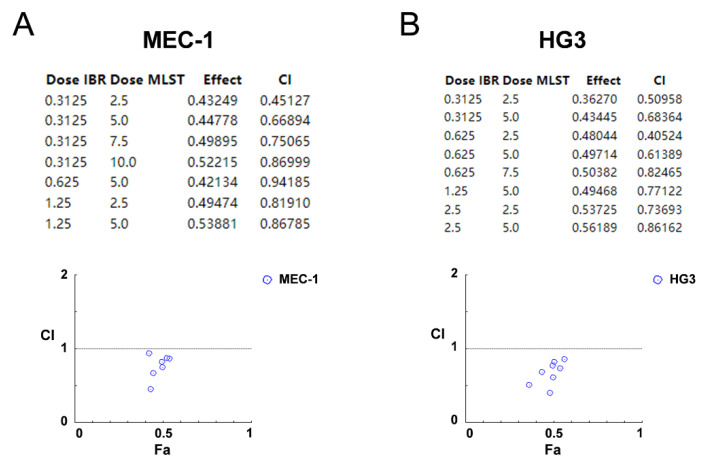
Synergistic effects of molidustat and ibrutinib. (**A**) Synergistic cytotoxicity of molidustat and ibrutinib on MEC-1 cells. The table shows the cell viability and synergistic index for each combination of molidustat and ibrutinib at different concentrations. The vertical axis in the dot plot is the cytotoxicity (CI) index, with values less than 1 indicating synergistic cytotoxicity of the two drugs at this concentration. (**B**) Synergistic cytotoxicity of molidustat and ibrutinib on HG3 cells. The table shows the cell viability and synergistic index for each combination of molidustat and ibrutinib at different concentrations. The vertical axis in the dot plot is the CI index, with values less than 1 indicating synergistic cytotoxicity of the two drugs at this concentration.

## Data Availability

Data are contained within the article and Appendix A.

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
