# Peer review of "HIF-PH Encoded by EGLN1 Is a Potential Therapeutic Target for Chronic Lymphocytic Leukemia"

_pharmaceuticals, 2022, doi:10.3390/ph15060734_

Round 1

Reviewer 1 Report

The authors describe treatment with molidustat as an alternative treatment in CLL with BTK resistence. The paper is put together well, the approach and technical work is solid, there are a few minor remarks.

1.       The figures are very small and the resolution is not great

2.       The introduction is long and the part with all the results could be removed.

3.       The first results 2.1 could use more explanation of what the comparisons are what normal cells are used, are they B cells?

4.       Figure 4: the description seems incorrect for Figure 4A, it says a high concentration of molidustat resulted in 30% survival, should be ibrutinib.

Author Response

Thank you for your valuable comments on this article. We have carefully revised the manuscript according to your requirements

     1. The figures are very small and the resolution is not great

Thanks for your attention. We have re integrated all the original images and adopted high-resolution images.

  1. The introduction is long and the part with all the results could be removed.

We have deleted the description of the result in the introduction section. In addition, we have added a figure with a schematic representation of the study, and replaced the last paragraph in introduction section with a short description of the conclusion of this paper (page3).

  1. The first results 2.1 could use more explanation of what the comparisons are what normal cells are used, are they B cells?

The control cells we used were peripheral blood mononuclear cells, which were separated by lymphocyte separation kit. The specific separation method has been described in method 4.11. They contain B cells, T cells and some other cells.

  1. Figure 4: the description seems incorrect for Figure 4A, it says a high concentration of molidustat resulted in 30% survival, should be ibrutinib.

Thank you for reading our work carefully. We have revised it (page 8).

We thank you again for your valuable comments on this article. If there are other problems, please contact us without hesitation

Reviewer 2 Report

The manuscript entitled “HIF-PH encoded by EGLN1 is a potential therapeutic target for chronic lymphocytic leukemia” by Guo et al, aimed to study the effect of molidustat in solid tumor leukemia. They identified differences in HIF1A expression between CLL patients and normal controls as well as an effect on the survival of CLL patients underling the potential of EGLN1 gene as a therapeutic target for CLL.

The work is well-organized and the results very interesting.

Some suggestions:

1. In the introduction, a figure containing the chemical structure of inhibitors and drugs should be added (Molidustat, venetoclax, idelalisib, ibrutinib..)

2. Figure 5B is not clear. Why are there three images of wester blot at and 20 µM? Please insert, the marker, at the height of the protein of interest.

3. I suggest including a figure with a schematic representation of the study.

Author Response

Thank you for your valuable comments on this article. We have carefully revised the manuscript according to your requirements

  1. 1.In the introduction, a figure containing the chemical structure of inhibitors and drugs should be added (Molidustat, venetoclax, idelalisib, ibrutinib..)

    Thanks for your advice. We have added chemical structures of these compounds in the introduction section (page2).

    2.Figure 5B is not clear. Why are there three images of wester blot at and 20 µM? Please insert, the marker, at the height of the protein of interest.

    Thanks for your question. As is shown in the figure note. Figure 5B is the results of HIF1α protein expression level in three primary CLL patients’ samples treated with 0 or 20 μM molidustat for 48 h. We put the figure there to illustrate the HIF1α protecting effect in primary CLL cells. We marked the molecular weight of the protein at the target band. In the process of Western blot, we cut the membrane and could not expose the marker in the development (page 9).

    1. I suggest including a figure with a schematic representation of the study.

    We have added a figure with a schematic representation of the study in the introduction part (page 3).

We thank you again for your valuable comments on this article. If there are other problems, please contact us without hesitation

Round 2

Reviewer 2 Report

The manuscript is now suitable for the publication